**Transported aerosols regulate the pre-monsoon rainfall over North-East India: a WRF-**
**Chem modelling study**
Neeldip Barman[1], Sharad Gokhale[2]
[1]Department of Civil Engineering, Indian Institute of Technology Guwahati, Guwahati, 781039, India
[2]Department of Civil Engineering, Indian Institute of Technology Guwahati, Guwahati, 781039, India
*Correspondence to*: Sharad Gokhale (sharadbg@iitg.ac.in)
**Abstract.** The study differentiates and quantifies the impacts of aerosols emitted locally within North-East (NE)
India region and those transported from outside this region to ascertain whether local or transported aerosols are
more impactful in influencing this region's rainfall during the pre-monsoon season (March-April-May). Due to
the existence of a declining pre-monsoon rainfall trend in NE India, the study also quantified the role of different
aerosol effects on radiative forcing (RF) and rainfall. The study has been carried out using the WRF-Chem model
by comparing simulation scenarios where emissions were turned on and off within and outside the NE region.
The impact of all emissions as a whole and Black carbon (BC) specifically was studied. Results show that aerosols
transported primarily from the Indo-Gangetic Plain (IGP) were responsible for 93.98 % of the $PM_{10}$ mass over
NE India's atmosphere and 64.18 % of near-surface $PM_{10}$ concentration. Transported aerosols contributed >50 %
of BC, organic carbon, sulfate, nitrate, ammonium and dust aerosol concentration and hence a major contributor
to air pollution. Hence, the aerosol effects were much greater with transported aerosols. Indirect aerosol effect
was found to be the major effect and more impactful with transported aerosols that dominated both rainfall and
RF, and suppressed rainfall significantly than the direct and semi-direct effect. However, the increase in direct
radiative effects with an increase in transported BC counteracted the rainfall suppression caused by relevant
processes of other aerosol effects. Thus, this study shows atmospheric transport to be an important process for
this region as transported emissions, specifically from IGP were also found to have greater control over the
region's rainfall. Thus, emission control policies implemented in IGP will reduce air pollution as well as the
climatic impacts of aerosols over the NE India region.

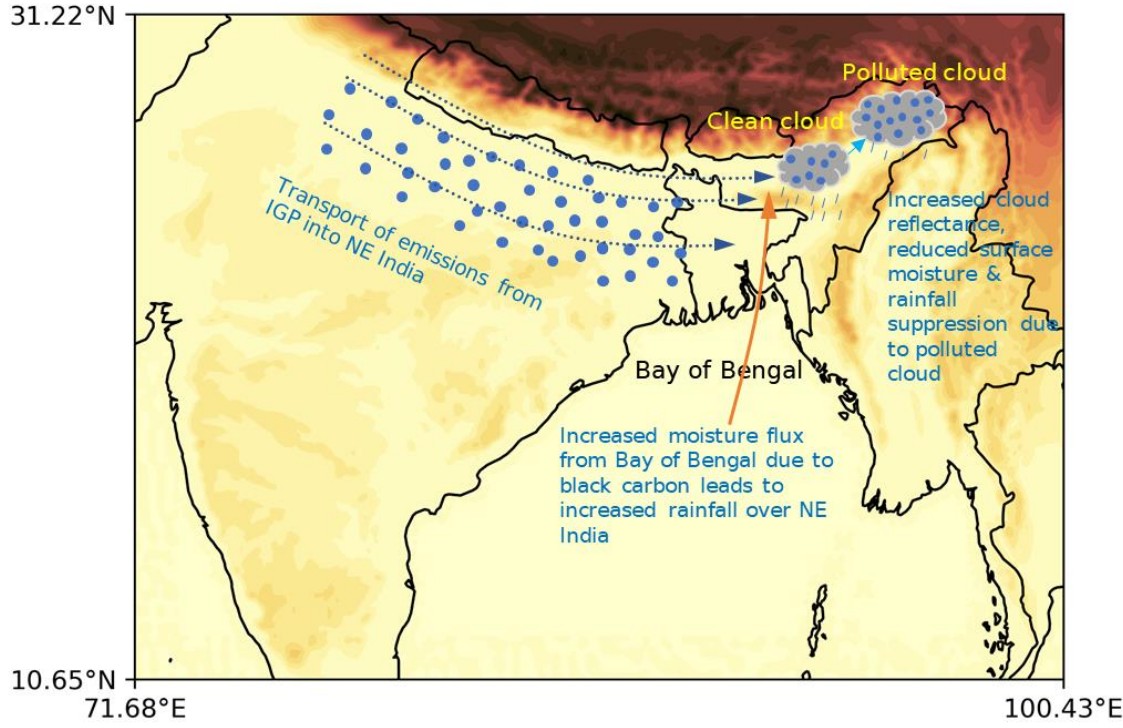

## 1 Introduction

Aerosols regulate the Earth's energy budget and hydrological cycle through scattering and absorption of solar
radiation and acting as sites for the formation of cloud droplets, which leads to its varied effects, viz. direct, semi-
direct and indirect effects (Mitchell, 1971; Rosenfeld, 2012; Menon et al., 2002). The effects differ spatially
depending on the constituents of aerosols, their physical and chemical properties as well as the quantity. Among
these factors, atmospheric transport also plays an important role which extends the climatic impacts to the
transported region from the source region (Lee et al., 2022). The IGP is a global hotspot of diverse aerosols (Ojha
et al., 2020; Kumar et al., 2018) that impacts regional and global climate (Ramanathan et al., 2005; Tripathi et al.,
2005; Sarangi et al., 2015). Air masses transport aerosols from the IGP to nearby regions, which also impact air
quality (Bhat et al., 2022; Ojha et al., 2012). Bonasoni et al. (2010) showed that pollutants from the IGP follow
the southern slope of the Himalayas as a path into the Bay of Bengal and NE India and similar observations were
made by Gogoi et al. (2017). The condition becomes more critical in the pre-monsoon season when the westerlies
directly transport air pollutants from the IGP to NE India. Among the aerosols, BC is a high climate-influencing
aerosol component due to its strong absorption capability (Bond et al., 2013; Nenes et al., 2002; Koch and Del
Genio, 2010) and IGP is the largest source region of it in India (Rana et al., 2019). Several studies (Guha et al.,
2015; Sarkar et al., 2019; Chatterjee et al., 2010) found BC, among other aerosols measured at sites in NE India
to be transported from the IGP. Moreover, in the NE India region, an increase in BC emissions was observed
along with high BC concentrations near the surface level (Barman and Gokhale, 2019; Chaudhury et al., 2022;
Singh and Gokhale, 2021). Tiwari et al. (2016) observed maximum BC concentration during this season in this
region along with the highest surface RF. The region also observes the highest atmospheric heating and highest
aerosol optical depth with an increasing trend during this period (Nair et al., 2017; Dahutia et al., 2018; Dahutia
et al., 2019; Gogoi et al., 2017; Pathak et al., 2010; Pathak et al., 2016). The presence of high aerosol loading
along with high atmospheric heating is likely to have varied aerosol effects over the region and may also have an
important role to play with the rainfall. Mondal et al. (2018) showed a decreasing trend of pre-monsoon rainfall
in this biodiversity hotspot region. Few modelling studies (Kant et al., 2021; Kedia et al., 2016; Kedia et al., 2019)
are available that studied the aerosol effect on rainfall over India. However, only Soni et al. (2017) and Barman
and Gokhale (2022) studied the BC effect on pre-monsoon rainfall in this region but without the inclusion of
aerosol indirect effect. Both studies found BC to increase total rainfall but Barman and Gokhale (2022) also found
semi-direct effect to be a rainfall suppression mechanism by evaporating clouds between 1 to 2 km above ground
level.
However, a few questions remained to be answered. How much is the contribution of transported aerosols
to air pollution and climatic effects compared to those emitted within NE India region? What is the role of different
aerosol effects on the rainfall mechanisms? Thus, this study was carried out with the following objectives (a)
Compare the contributions of local and transported aerosols to air pollution and different climatic effects over NE
India (b) Quantify the role of different aerosol effects on the climatic effects (c) Investigate the role of BC emitted
within NE India and transported BC in such climatic effects. Here, transported aerosols include the transported
primary aerosols emitted from outside NE India as well as the secondary aerosols formed from the transported
emissions. Same goes for local emissions. Through qualitative and quantitative comparison of the impacts of local
and transported aerosols, the study tries to find the source region of aerosols that has a greater impact on the
atmosphere over NE India during the pre-monsoon season. Since observational studies cannot distinguish between
the local and transported aerosols impacts, the study was carried out with numerical modelling. The effect of
transported aerosols on different regions of the world has been studied (Krishnamohan et al., 2021; Wang et al.,
2020; Bagtasa et al., 2019) but none of them covered the IGP and its impact on the nearby region.
**2 Methods**
The study used the WRF-Chem v4.2.1 model (Grell et al., 2005). The model configuration, modelling domain,
model inputs and simulation period is similar to the one used in Barman and Gokhale (2022). Details regarding
physical and chemical parametrization schemes and the emissions are provided in Table 1.
Table 1: Details of physical parametrizations, chemical parametrizations and emissions

| **Physical parametrizations** | |
|---|---|
| Planetary boundary layer | MYNN3 (Nakanishi and Niino, 2006) |
| Radiation | RRTMG (Iacono et al., 2008) |
| Land surface model | NOAH (Tewari et al., 2004) |
| Cumulus scheme | Grell-Freitas (Grell and Freitas, 2014) |
| Microphysics | Morrison (Morrison et al., 2009) |
| Meteorology initial and boundary conditions | ERA5 (Hersbach et al., 2020) |
| **Chemical parametrizations and emissions** | |
| Chemistry scheme | MOZART (Emmons et al., 2010) |
| Aerosol scheme | MOSAIC (Zaveri et al., 2008) |
| Chemistry initial and boundary conditions | CAM-Chem (Lamarque et al., 2012) |
| Anthropogenic emissions | CAMS emission inventory (Granier et al., 2019) |
| Fire emissions | FINN (Wiedinmyer et al., 2010) |

| | | |
|---|---|---|
| Dust emissions | Online model (Zhao et al., 2010) | |
| Biogenic emissions | MEGAN v2.04 (Guenther et al., 2006) | |


75        The model was run at 10 km grid size for a duration of 13 days from 7-19 April 2018, out of which a 3-
day period from 7-9 April 2018 was discarded as spin-up and outputs from 10-19 April 2018 were used for
analysis. The period represents the mid of pre-monsoon season. Also, April 2018 was Indian Ocean Dipole and
ENSO neutral period and hence suitable for study of aerosol effects. The model domain is shown in Fig. 1(a)
which extends from 10.65° N to 31.22° N and 71.68° E to 100.43° E, and the NE India is the part of India within
the region bounded by the blue box. The region within the box is bounded by 22° N and 29° N latitudes and 89°
E and 97° E longitudes. The climatic situation during the study period was also described in Barman and Gokhale
(2022). The near surface wind flow was from the Bay of Bengal towards NE India, which gradually changed to
westerly wind flow carrying aerosols from IGP towards NE India. Hence the domain was selected by keeping the
NE India region near the upper-right corner of the domain. Descriptions of the simulations are provided in Table
85 2.

Table 2: Description of simulations

| | Simulation name | Description of simulations |
|---|---|---|
| 1. | NOR-I | Baseline simulation with all aerosol effects |
| 2. | NOFEED-I | Same as NOR-I but with aerosol radiative effects turned off |
| 3. | NOCHEM | Simulation with no atmospheric chemistry and aerosol effects |
| 4. | No_EMISS_NE | Same as NOR-I but with emissions turned on only outside NE India |
| 5. | Only_EMISS_NE | Same as NOR-I but with emissions turned on only within NE India |
| 6. | No_EMISS_NE_4SO$_2$ | Same as No_EMISS_NE but with 4×SO$_2$ emissions |
| 7. | No_EMISS_NE_0.25SO$_2$ | Same as No_EMISS_NE but with 0.25×SO$_2$ emissions |
| 8. | No_EMISS_NE_NOFEED | Same as No_EMISS_NE but with aerosol radiative effects turned off |
| 9. | Only_EMISS_NE_NOFEED | Same as Only_EMISS_NE but with aerosol radiative effects turned off |
| 10. | No_NE_BCI | Same as NOR-I but with BC emissions turned on only outside NE India |
| 11. | Only_NE_BCI | Same as NOR-I but with BC emissions turned on only within NE India |
| 12. | 4NOR-I | Same as NOR-I but with 4×BC emissions |
| 13. | No_BC_ABS | Same as NOR-I but with BC absorption disabled |
| 14. | NOR | Baseline simulation with only direct and semi-direct effect |
| 15. | 2NOR | Same as NOR but with 2×BC emissions |
| 16. | No_NE_BC | Same as NOR but with BC emissions within NE India region turned off |
| 17. | No_NE_2×BC | Same as No_NE_BC but with 2×BC emissions outside NE India |
| 18. | Only_NE_BC | Same as NOR but with BC emissions turned off outside NE India |
| 19. | Only_NE_2×BC | Same as Only_NE_BC but with 2×BC emissions inside NE India |
| 20. | NOFEED | Same as NOR but with aerosol radiative effects off |


All the simulations were conducted with the MOZART-MOSAIC scheme, except simulation 3, which was purely a meteorology simulation and did not include any atmospheric chemistry and aerosol effects. Moreover, simulations 1 to 13 (except 3), were conducted with the version of MOZART-MOSAIC scheme which also supports indirect aerosol effect by coupling with the Morrison microphysics scheme along with direct and semi-direct effect, while simulations 14 to 20 did not include indirect effect. The NOR simulation used in Barman and Gokhale (2022), was also used in this study. NOR-I is also the baseline simulation run with the same baseline emissions for the study period as NOR, but also includes indirect aerosol effect. No_EMISS_NE had all emissions (biogenic, anthropogenic and dust) disabled within the region bounded by 22° N and 29° N latitudes and 89° E and 97° E longitudes, shown by the blue box in Figure 1(a) while No_NE_BC and No_NE_BCI only had BC emissions disabled within the same region. Only_EMISS_NE had all emissions disabled outside of the above region along with boundary conditions for all chemical species modified to zero to nullify the transport of emissions from outside the domain and similarly, Only_NE_BC and Only_NE_BCI had BC emissions disabled outside the NE India region with boundary conditions for BC modified to zero. Remaining simulations can be understood from Table 2 and their applications are understood from the results and discussion in Sect 3.

As per Ghan et al. (2012) and Bauer and Menon (2012), the total aerosol effect is the algebraic sum of direct, indirect and semi-direct effects. Similar approaches were used by Yang et al. (2011). Thus,

NOR-I – NOCHEM = Total aerosol effect = Direct + Semi-direct + Indirect,     (1)

Both NOFEED-I and NOR-I includes indirect effect but NOFEED-I does not include aerosol radiative effects. Thus,

NOR-I – NOFEED-I = Direct + Semi-direct effect,     (2)

Also, since NOFEED-I includes only indirect effect,

NOFEED-I – NOCHEM = Indirect effect,     (3)

Similar approaches were used by Wang et al. (2015).

The NOR simulation utilised in this study was evaluated in Barman and Gokhale (2022). Moreover, meteorological evaluation of NOR-I w.r.t wind direction, wind speed, temperature and humidity was carried out against surface station datasets (https://mesonet.agron.iastate.edu/sites/locate.php) at Guwahati (26.10 °N, 91.58 °E), Kolkata (22.65 °N, 88.45 °E), Bangalore (13.20 °N, 77.70 °E), Patna (25.59 °N, 85.08 °E), Delhi (28.56 °N, 77.11 °E) and Mumbai (19.10 °N, 72.86 °E). Simulated rainfall was evaluated against the Indian Meteorological Department (IMD) rainfall dataset of Pai et al. (2014) (https://www.imdpune.gov.in/Clim_Pred_LRF_New/Grided_Data_Download.html). Index of agreement (IOA), root mean square error (RMSE) and mean error (ME) were used as statistical parameters. As per the criteria of Emery et al. (2001), the NOR-I simulation underpredicted temperature but showed good performance with wind speed and wind direction but had large RMSE with wind direction, similar to the NOR simulation. Performance statistics are provided in Table S1. Moreover, NOR and NOR-I simulated chemical species (BC, organic carbon, dust and sulfate aerosol) were compared against the MERRA2 dataset (https://disc.gsfc.nasa.gov/datasets/M2T1NXAER_5.12.4/summary) at the above locations. Performance statistics are shown in Table S2. NOR gave a much better estimation of all the chemical species at all locations.

Moreover, the predicted chemical species of nitric oxide (NO), nitrogen dioxide (NO$_2$), sulfur dioxide (SO$_2$),
PM$_{2.5}$ and PM$_{10}$ were compared against in-situ observations at Delhi (28.56 °N, 77.11 °E), Kanpur (26.57 °N,
80.32 °E), Patna (25.61 °N, 85.13 °E) and Siliguri (26.69 °N, 88.41 °E), obtained from Central Pollution Control
Board, India (https://app.cpcbccr.com/ccr/#/caaqm-dashboard-all/caaqm-landing/caaqm-data-availability). These
locations are located along the IGP. Performance statistics are given in Table S3. The performance statistics were
better with both particulate matter than gaseous species. Comparatively the performance was better with
MERRA2. The relatively lower performance with in-situ comparison may be due to the grid size as in-situ
observations are affected by local emission sources as well the deficiencies in emission inventory. However, the
inclusion of all aerosol effects greatly improved simulated rainfall performance with NE India regional average
IOA: 0.52, ME: 3.72 mm day$^{-1}$, RMSE: 13.55 mm day$^{-1}$ compared to only considering direct + semi-direct effect
(IOA: 0.40, ME: 9.22 mm day$^{-1}$, RMSE: 21.26 mm day$^{-1}$) in Barman and Gokhale (2022). The improvement in
performance and decrease in ME show that indirect effect played a major role during this period in controlling
and suppressing rainfall.
**3 Results and discussion**
**3.1 PM$_{10}$ spatial and vertical distribution**

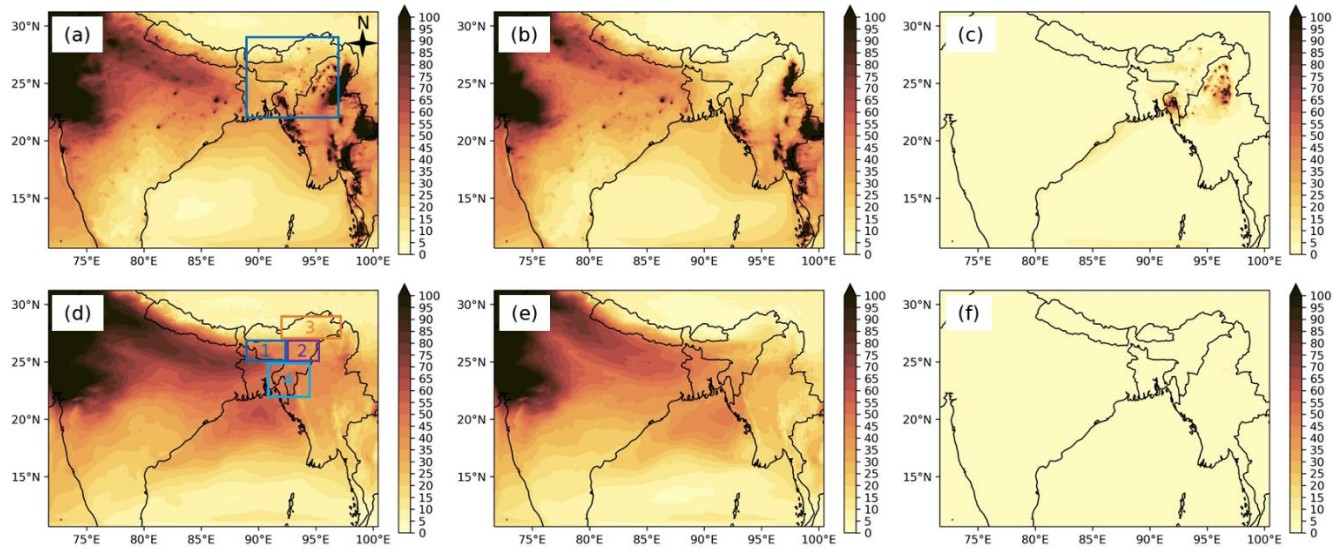


Figure 1: Spatial distribution of PM$_{10}$ concentration (μg m$^{-3}$) in NOR-I, (a, d), No_EMISS_NE (b, e) and
Only_EMISS_NE (c, f). Upper row shows distribution at model level 0 (near surface) and the lower row at model
level 15
Figure 1 shows the time-averaged spatial distribution of PM$_{10}$ concentration. The NE India region was divided
into four regions based on the proximity from the IGP, shown in Fig. 1(d). Region 1 and region 2 fall along the
Brahmaputra River Valley, with region 1 being closest to IGP. Region 3 is mostly a mountainous region and 4 is
the southern region closer to the Bay of Bengal. The spatial distribution of geopotential heights of model level 0
and 15 are shown in Fig. S1, while region-wise (Fig. 1(d)) concentration values within NE India at the two
atmospheric heights are shown in Table S4. PM$_{10}$ concentration contours shown in Fig. 1(a), 1(b), 1(d) and 1(e),
emanating from IGP and spreading into NE India indicated the transport of aerosols from IGP into NE India. The
similarity of these spatial distributions of No_EMISS_NE to the baseline scenario, NOR-I, especially within NE
India region inferred that most of the aerosol mass within NE India was contributed by transported aerosols, while
$PM_{10}$ emitted or formed over NE India remained mainly confined within the region as shown in Fig. 1(c), possibly
due to the mountainous terrain, as also described in Kundu et al. (2018). The transport of $PM_{10}$ can also be seen
from Fig. 2, in which the streamline's arrow from IGP to NE India show the transport of air-mass and the colour
of the streamlines show the $PM_{10}$ mass flux in $\mu g\ m^{-2}\ s^{-1}$. The flux was higher over IGP.

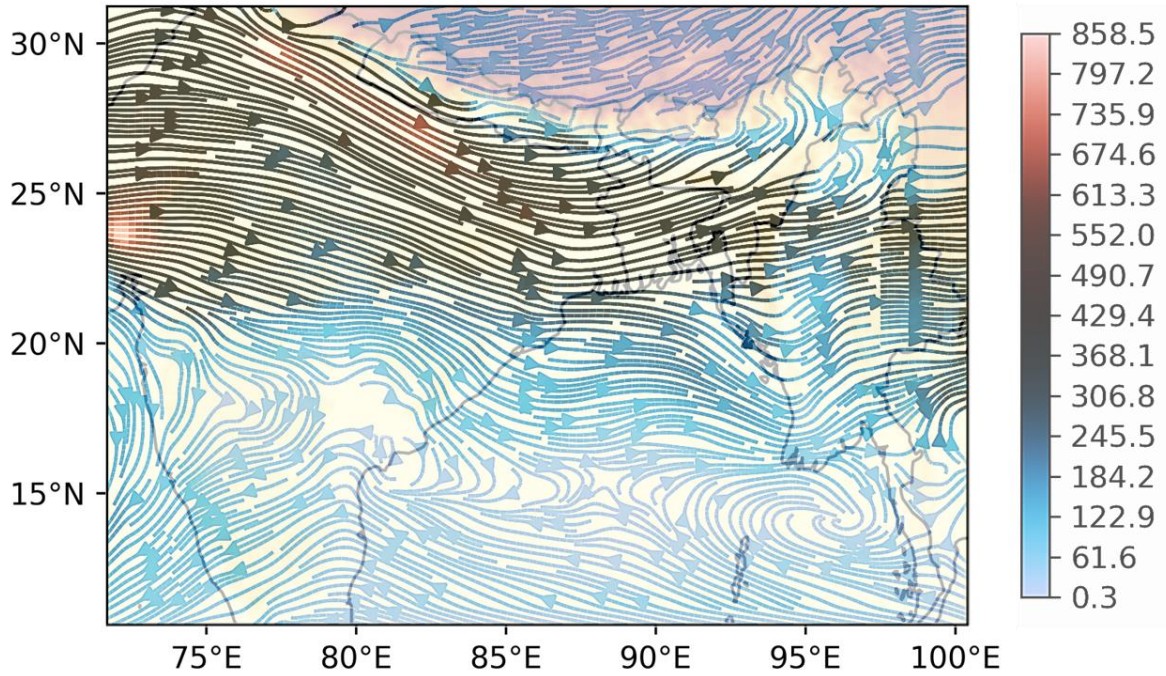


Figure 2: Streamlines showing transport of air-mass from IGP to NE India and $PM_{10}$ mass flux ($\mu g\ m^{-2}\ s^{-1}$) at 1300
m above terrain
Both near the surface and at higher atmosphere (level 15), No_EMISS_NE showed a higher regional average
concentration (surface: 14.46 $\mu g\ m^{-3}$, higher atmosphere: 24.43 $\mu g\ m^{-3}$) which was closer to the baseline scenario
of NOR-I (surface: 27.43 $\mu g\ m^{-3}$, higher atmosphere: 34.13 $\mu g\ m^{-3}$) compared to the local emission scenario of
Only_EMISS_NE (surface: 8.07 $\mu g\ m^{-3}$, higher atmosphere: 0.98 $\mu g\ m^{-3}$). Thus, transported aerosols contributed
higher $PM_{10}$ concentration (64.18 %) than local emission and contribution from local emissions were negligible
at higher atmosphere, as also seen in Fig. 1(f) and 96.14 % of it was contributed by transported aerosols. The
higher concentration at higher atmosphere was due to transported aerosols developing an elevated $PM_{10}$ profile
(Fig. S2) having maximum concentration near 2000 m and which shows much greater similarity with the baseline
scenario. The long range transport and strong convective active during this season is responsible for the elevated
profile (Pathak et al.. 2016). Hence, transported aerosols contributed to bulk of the aerosols over NE India
throughout the atmospheric column (93.98 %) indicated by the column integrated $PM_{10}$ mass of 313.97 g $m^{-2}$
(No_EMISS_NE) and 20.08 g $m^{-2}$ (Only_EMISS_NE). NOR-I had column integrated $PM_{10}$ mass of 466.63 g $m^{-}$
$^{2}$. Further analysis indicated that transported aerosols accounted for >50 % of BC, organic carbon, sulfate, nitrate,
ammonium and dust aerosol mass over NE India's atmosphere as the column integrated mass for these species in
No_EMISS_NE were 4.55, 19.59, 51.66, 2.20, 13.74 and 207.82 g $m^{-2}$, respectively, while it was 0.94, 6.51, 1.79,
0.12, 0.56 and 6.60 g m$^{-2}$ , respectively in Only_EMISS_NE. The spatial distribution of column integrated mass
of these species can be seen in Figures S3, S4, S5, S6, S7 and S8. Regions 1, being in close proximity to IGP, as
seen in Fig. 1(c)), received maximum near surface aerosol mass (73.70 %) from transported aerosols, compared
to the other regions, followed by region 2 (66.86 %), 3 (60.48 %) and 4 (57.43 %). However, even though
No_EMISS_NE and Only_EMISS_NE is the bifurcation of NOR-I into two separate emission regions, the sum
of No_EMISS_NE and Only_EMISS_NE column integrated mass as well as concentrations didn't equate to NOR-
I values and is always less than it. This indicated formation of extra aerosol mass due to interaction of emissions
of the two regions.
**3.2 Aerosol effects of local and transported aerosols on radiative forcing**
RF due to different aerosol effects was estimated based on the methodology described in Sect. 2. Further details
regarding its estimation are provided in the supplementary.

186        The baseline scenario indicated that direct and indirect aerosol effects caused net (NET) surface and top

of the atmosphere (TOA) dimming while causing atmospheric heating, as seen in Fig. 3. This is due to the presence
of aerosols that scatter and absorb solar radiation, reducing it at the surface while increasing it at the top of the
atmosphere as well as causing atmospheric heating. Net direct surface, TOA and atmospheric RF were -15.34, -
7.49 and 7.85 Wm$^{-2}$ and was mainly contributed by short-wave (SW) radiation. Indirect effect had the same effect
on solar radiation as the direct effect and was due to the formation of numerous smaller cloud droplets which has
better reflectivity to solar radiation, also known as the 1$^{st}$ indirect effect or Twomey effect (Twomey, 1977).
However, positive atmospheric RF (18.20 W m$^{-2}$) causing atmospheric heating (10.06 W m$^{-2}$) was mainly caused
by long-wave (LW) radiation (16.22 W m$^{-2}$) at the TOA contributed by indirect effect. This was due to greater
cloud cover (Fig. S9) at 8 – 10 km which is not seen in the other two scenarios. The indirect effect also caused
warming at the surface (6.17 W m$^{-2}$), as its contributed to greater cloud cover (Nandan et al., 2022) and caused
heating of the surface through LW radiation. The total net surface RF was -27.88 W m$^{-2}$ out of which -23.92 W
m$^{-2}$ or 85.80% was contributed

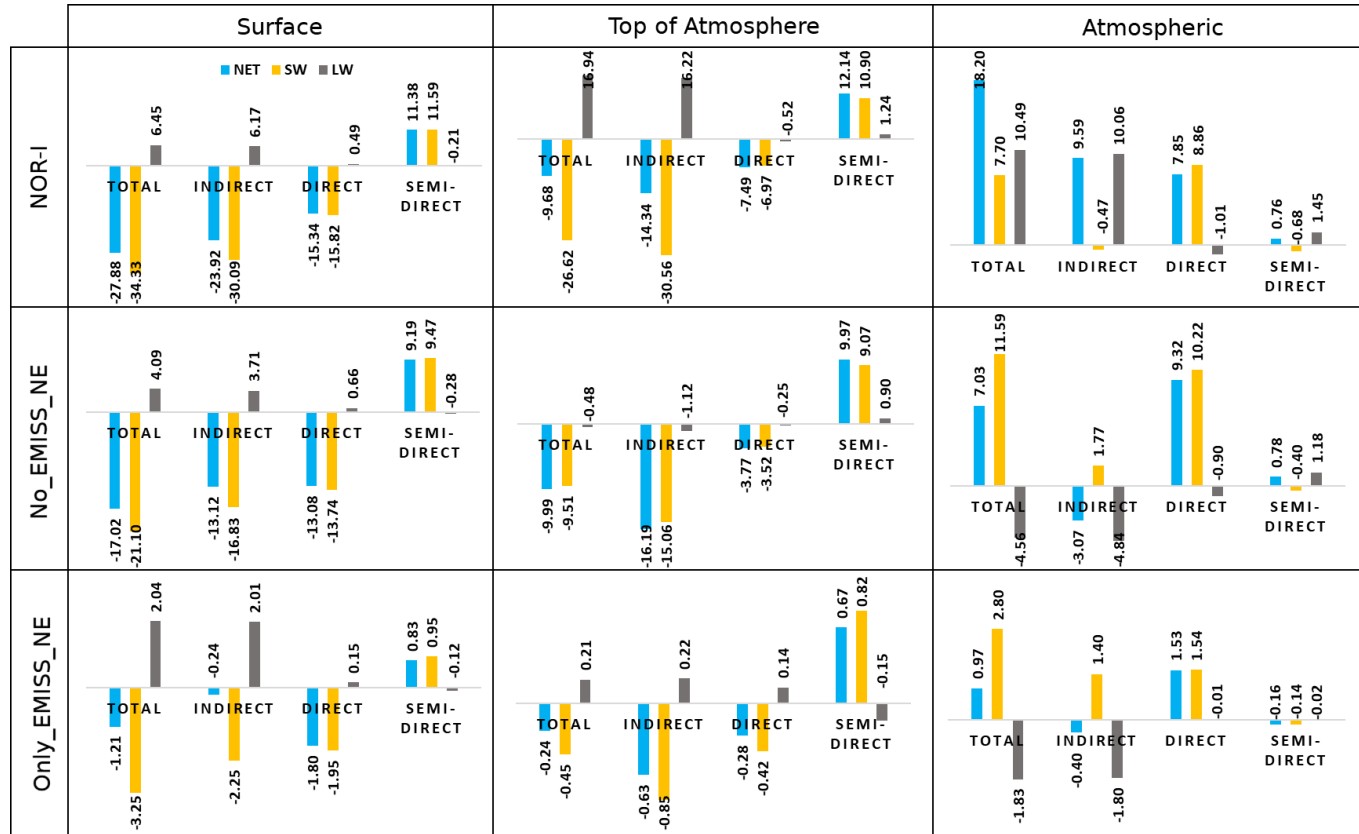

Figure 3: NE India regional average RF (W m⁻²) due to different aerosol effects at NET, SW and LW wavelengths in different emission scenarios

by indirect forcing. Indirect SW forcing (-30.08 W m⁻²) was almost twice the direct SW forcing (-15.82 W m⁻²), while semi-direct SW forcing (+11.58 W m⁻²) was ~75% of the direct forcing. Semi-direct effect showed positive surface RF due to cloud cover reduction. Thus, atmospheric heating and the subsequent evaporation of clouds compensated to a large extent the reduction in solar radiation due to aerosols. The atmospheric RF (0.76 W m⁻²) due to semi-direct effect was due to LW radiation, which may be due to increased solar radiation at the surface, which released the heat into the atmosphere in the form of LW radiation. However, this value was very small. The indirect RF contributed most to the total surface, TOA and atmospheric RF at both SW and LW wavelengths and hence was found to be the dominant aerosol effect affecting radiation over NE India.

Quantitatively, No_EMISS_NE provided RF values (surface: -17.02 W m⁻², TOA: -9.99 W m⁻² and atmospheric RF: 7.03 W m⁻²) that were much similar and closer to the baseline scenario (surface: -27.88 W m⁻², TOA: -9.68 W m⁻² and atmospheric RF: 18.20 W m⁻²) than Only_EMISS_NE (surface: -1.21 W m⁻², TOA: -0.24 W m⁻² and atmospheric RF: 0.97 W m⁻²). Consequently, the No_EMISS_NE net indirect, direct and semi-direct surface RF values of -13.12, -13.08 and 9.19 W m⁻² were significantly larger than the corresponding Only_EMISS_NE RF values of -0.24, -1.80 and 0.83 W m⁻². A similar conclusion could be inferred at TOA also. Hence transported aerosols were primarily responsible for all the different aerosol effects on radiation over NE India as a greater amount of aerosol mass was contributed by it. Moreover, No_EMISS_NE net direct atmospheric RF (9.32 W m⁻²) was found to be even higher than the baseline scenario (7.85 W m⁻²). This indicated that the NE India region contained more scattering aerosols while transported aerosols contained more absorbing aerosols as

the difference in the direct atmospheric RF is mainly driven by changes in the TOA RF (-7.49 vs. -3.77 W m$^{-2}$) than surface RF (-15.34 vs. -13.08 W m$^{-2}$). Region 1 had the highest direct and semi-direct net surface RF of -20.41 W m$^{-2}$ and 19.20 W m$^{-2}$, respectively due to its close proximity to IGP.

**3.3 Aerosol effects of local and transported aerosols on rainfall**

Table 3: Changes in rainfall due to different aerosol effects in different scenarios (mm)

|  | Total aerosol effect | Direct + semi-direct | Indirect |
|---|---|---|---|
| **NOR-I** | -275.13 | -17.04 | -258.09 |
| **No_EMISS_NE** | -73.06 | -23.95 | -49.11 |
| **Only_EMISS_NE** | -24.45 | -8.42 | -16.04 |

The quantitative changes in regional average rainfall amounts over NE India due to the different aerosol effects induced by the aerosols in different scenarios are provided in Table 3. Region-wise values can be read from Table S5. Rainfall from region 4 was not considered due to large errors being associated with it (Fig. S10). In the baseline scenario (NOR-I), the total aerosol effect caused rainfall suppression in all three regions, with a regional total of -275.13 mm, shown in Table 3. Reductions in rainfall due to the total aerosol effect was contributed by suppressions due to both direct + semi-direct and indirect effect and was observed in all the considered regions. The highest suppression was observed in region 3 (-102.60 mm), followed by region 1 (-100.60 mm). The role of direct + semi-direct effect was observed to be minimal with a total regional suppression of -17.04 mm while the indirect effect (-258.09 mm) was responsible for almost the whole of the suppression of -275.13 mm. Region 1 observed the highest suppression of -13.21 mm due to direct + semi-direct effect as this region's radiation was highest impacted by these effects.

Direct effect could suppress rainfall by reducing surface evaporation and convection through surface dimming while semi-direct by evaporation of clouds (Talukdar et al., 2019; Lohmann and Feichter, 2001; Habib et al., 2006; Bollasina et al., 2011; Koch and Del Genio, 2010b). However, the surface dimming by indirect effect (-23.92 W m$^{-2}$) with NOR-I was much larger than the combined direct + semi-direct effect (-3.96 W m$^{-2}$). Hence the reduction in surface moisture flux due to indirect effect (-6.45×10$^{-6}$ kg m$^{-2}$ s$^{-1}$) was much greater than due to combined direct + semi-direct effect (-1.1×10$^{-6}$ kg m$^{-2}$ s$^{-1}$) and much similar to the reduction due to total aerosol effect (-7.56×10$^{-6}$ kg m$^{-2}$ s$^{-1}$). This was also observed in the case of No_EMISS_NE. The greater surface dimming of -17.02 W m$^{-2}$ in No_EMISS_NE caused a much higher negative surface moisture flux change of -3.82×10$^{-6}$ kg m$^{-2}$ s$^{-1}$ due to total aerosol effect, mostly contributed by indirect effect (-2.79×10$^{-6}$ kg m$^{-2}$ s$^{-1}$) compared to direct + semi-direct effect (-1.03×10$^{-6}$ kg m$^{-2}$ s$^{-1}$). Hence, indirect effect in NOR-I and No_EMISS_NE dominated moisture reduction through reduction in surface moisture flux over most areas of NE India at both low and high-terrain regions, as seen in Fig. 4.

However, direct + semi-direct effect caused an increase of moisture in NOR-I and No_EMISS_NE over most of NE India in spite of a negative surface moisture flux not observed in Only_EMISS_NE. This indicated that direct + semi-direct caused an increase in the transport of moisture from another region, in this case from Bay of Bengal. The equivalent potential temperature (EPT) profiles in Fig. 5 compared the atmospheric stability due to

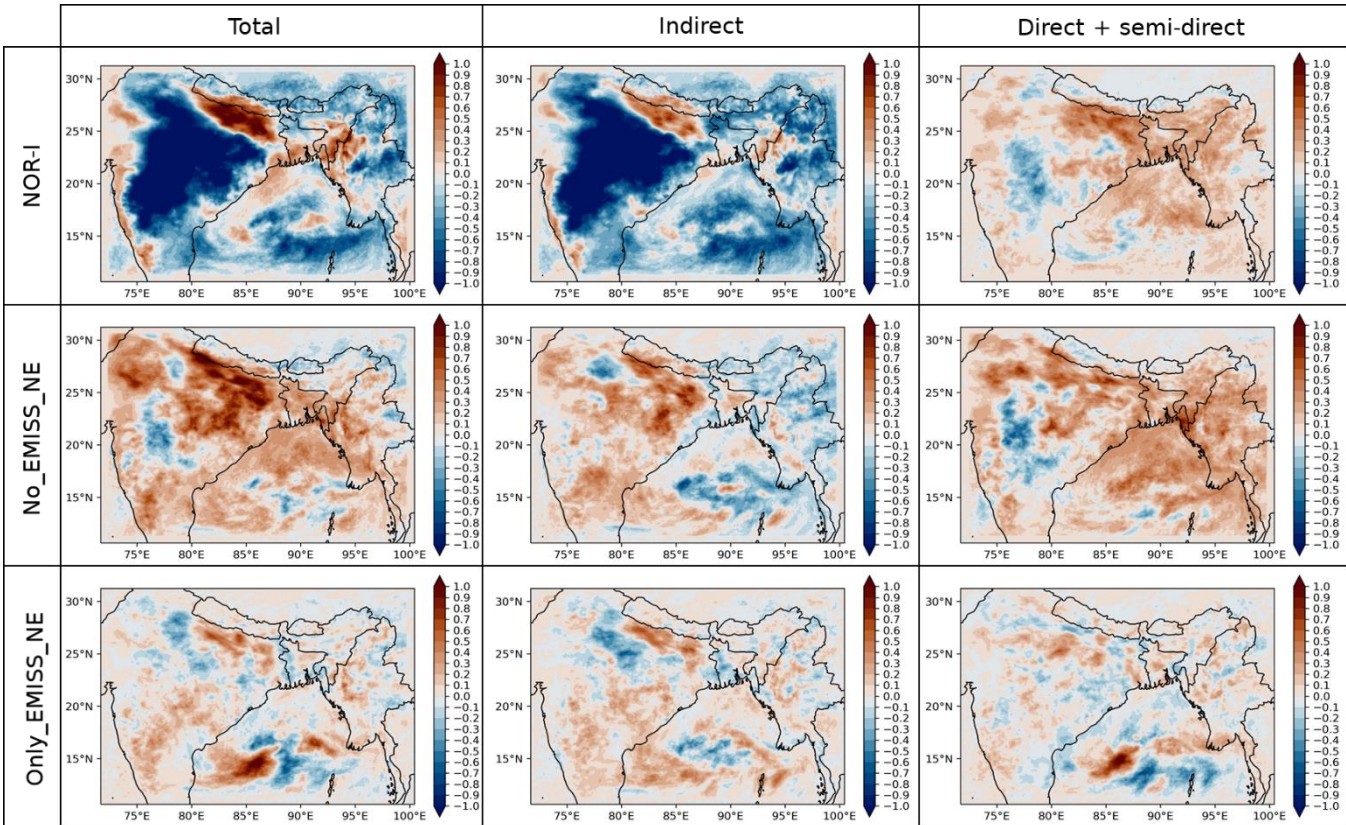

Figure 4: Spatial distribution of change in near-surface water vapor mixing ratio (g kg$^{-1}$) due to total aerosol effect, direct + semi-direct effect and indirect effect

different aerosol effects. The greater surface dimming due to the indirect effect in No_EMISS_NE caused not only negative surface moisture flux but also a significant increase in atmospheric stability (indicated by increasing value of indirect effect EPT profile with height), reducing convection, which possibly also contributed reduction to rainfall suppression. However, although the direct + semi-direct EPT profile showed increased atmospheric stability below 1 km, but created an overall unstable atmosphere in the lower atmosphere. This instability, primarily caused due to atmospheric heating of BC, created an unstable region over NE India which facilitated the increased transport of moisture from the Bay of Bengal (discussed later). Hence, though the direct effect reduces rainfall by reducing surface moisture flux and convection but also possibly enhances it by transporting moisture. This transported moisture possibly compensated to some extent the rainfall reduction due to a decrease in surface moisture flux, convection and cloud evaporation caused by direct and semi-direct effects. Hence, the rainfall reduction due to direct + semi-direct effect (-17.04 mm) was possibly significantly less than the indirect effect (-258.09 mm). Thus, the effect of direct and indirect effects on dynamics was distinctly different. The EPT profile of the total aerosol effect in No_EMISS_NE showed an unstable lower atmosphere, supporting moisture transport. Similar explanation could be given for moisture increase due to direct + semi-direct in NOR-I but the increase in atmospheric stability and moisture reduction due to greater surface dimming by its indirect effect was significantly larger, which created an overall stable atmosphere due to total aerosol effect in NOR-I. The EPT profiles of Only_EMISS_NE showed almost zero perturbation throughout the atmosphere and hence was unable to affect atmospheric stability and cause moisture transport. Thus, the direct + semi-direct effect in Only_EMISS_NE did not show significant moisture change in Fig. 4. Moreover, the significantly smaller surface dimming (-1.21 W m$^{-2}$) in Only_EMISS_NE caused very small but positive change of $8.15 \times 10^{-8}$ kg m$^{-2}$ s$^{-1}$ due to

the total aerosol effect and hence similar moisture change is observed in Fig. 4. Hence aerosols emitted solely
from NE India had negligible capability in affecting moisture through different aerosol effects. Moisture reduction
over NE India was much greater due to the indirect effect in No_EMISS_NE compared to Only_EMISS_NE,
while moisture increase was much greater in No_EMISS_NE compared to Only_EMISS_NE due to a higher
direct + semi-direct effect.

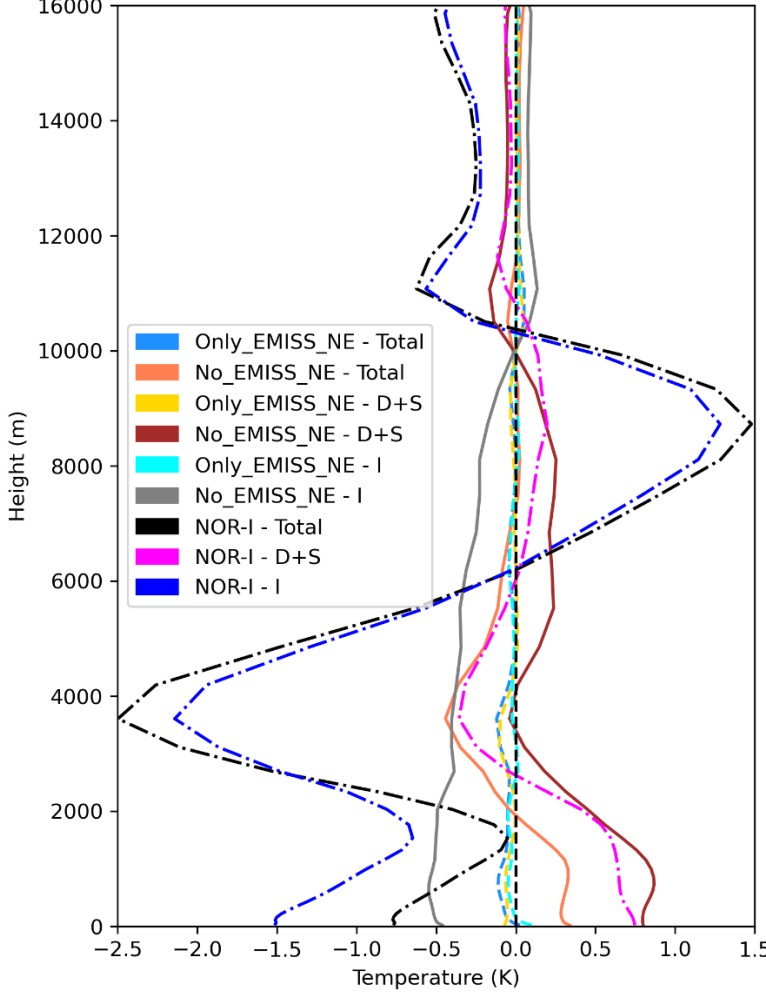


Figure 5: Perturbation of EPT (K) due to total aerosol effect (Total), direct + semi-direct (D+S) and indirect (I) aerosol effect
in No_EMISS_NE (non-dashed), Only_EMISS_NE (dashed) and NOR-I (dashdot)

285        Moreover, the positive NE India regional average difference of column integrated cloud condensation

nuclei (CCN) number ($4.38 \times 10^{10}$ m$^{-2}$), cloud droplet number ($4.42 \times 10^{13}$ m$^{-2}$) and cloudwater (27.93 g m$^{-2}$), and
estimated from No_EMISS_NE – Only_EMISS_NE indicated that transported aerosols had a greater impact
through aerosol indirect effect (Zhang et al., 2010). The presence of larger aerosol amounts in the form of CCN
affects the cloud lifetime by affecting the conversion from cloudwater to rainwater, thus, to rainfall, thereby
suppressing rainfall, also known as the 2nd indirect effect (Shiogama et al., 2010; Cherian et al., 2017). The
presence of a large amount of CCN facilitates condensation of water vapor on numerous CCN particles, producing
numerous cloud droplets with smaller radii. This restricts small cloud droplets to grow in size due to reduction in
interaction with other cloud droplets which affect its conversion to rain droplet, and thus to rainfall. Due to more
aerosol mass over NE India (Sect. 3.1), NOR-I and No_EMISS_NE had significantly higher cloudwater compared
to

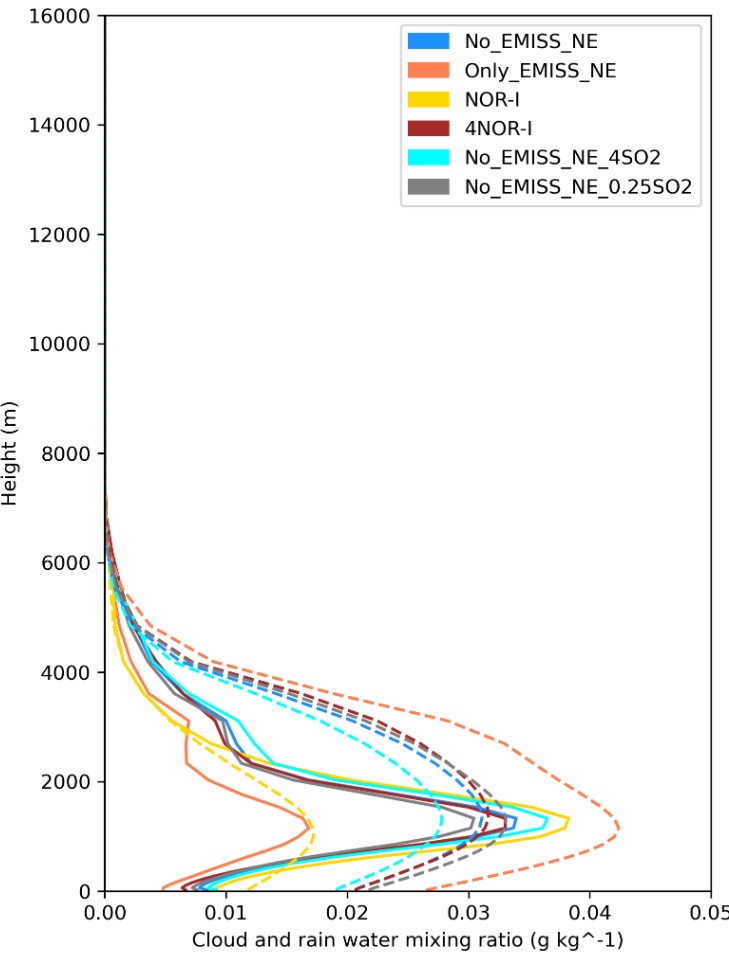

Figure 6: NE India regional average vertical profiles of cloudwater mixing ratio (non-dashed) and rainwater mixing ratio
(dashed) in different scenarios (g kg$^{-1}$)
Only_EMISS_NE, as seen in Fig. 6. Consequently, NOR-I and No_EMISS_NE had a significantly lower
rainwater mixing ratio than Only_EMISS_NE. Thus, rainfall suppression due to indirect effect was highest in
NOR-I, followed by No_EMISS_NE and Only_EMISS_NE. Hence, the combined effect of reduction in moisture,
instability and rainfall formation contributed to the reduction in rainfall through indirect and total aerosol effects.
This could be a possible key mechanism associated with the decreasing rainfall trend in the region. Reduction of
moisture due to the direct effect of aerosols and evaporation of clouds by BC were found to be possible
mechanisms by Barman and Gokhale (2022). However, this study shows that the contribution of direct and semi-
direct effects was very small compared to the indirect effect. The indirect effect has been found to be the dominant
aerosol effect in many studies (Wang et al., 2015; Liu et al., 2016) and was found to suppress monsoon rainfall
over India (Manoj et al., 2012). Aerosol indirect effect is mainly dictated by the warm clouds (Christensen et al.,
2016). Thus, the higher cloud cover associated with NOR-I and No_EMISS_NE in lower atmosphere which
affected SW radiation more in Sect. 3.2, was due to a greater amount of cloudwater in lower atmosphere.

311          Moreover, No_EMISS_NE and Only_EMISS_NE simulations were evaluated against the IMD rainfall

dataset and NOR-I simulation to check whether the local or transported aerosols had greater control over the

rainfall in NE India. No_EMISS_NE showed better regional average rainfall statistics than Only_EMISS_NE with higher IOA (0.48 vs. 0.47), lower RMSE (18.85 vs. 20.37 mm day$^{-1}$), and lower ME (6.94 vs. 8.22 mm day$^{-1}$) on comparing with the IMD rainfall dataset. Also, the simulated rainfall of No_EMISS_NE showed higher rainfall similarity with NOR-I than Only_EMISS_NE with higher IOA (0.65 vs. 0.63), lower RMSE (56.32 vs. 61.92 mm day$^{-1}$) and lower ME (39.30 vs. 39.81 mm day$^{-1}$). Hence, No_EMISS_NE showed more similarity with the baseline scenario as well as observed data and had greater control over the region's rainfall.

**3.4 Role of local and transported BC**

In section 3.3, the direct effect showed to increase moisture over NE India through an increase in atmospheric instability, caused mainly due to atmospheric heating of BC (Barman and Gokhale (2022)) Hence, to negate the effects of the indirect effect on atmospheric dynamics, scenarios in Table 1 containing only direct and semi-direct effects were used in this analysis. Moreover, NOR gave a much better performance with BC concentration estimation (Table S2) than when the indirect effect was included (NOR-I). The results from No_EMISS_NE, Only_EMISS_NE, No_NE_BCI and Only_NE_BCI scenarios were compared and related.

**3.4.1 Radiative heating**

The regional average vertical profiles of NOR, 2NOR, No_NE_BC, No_NE_2×BC, Only_NE_BC and Only_NE_2×BC can be seen from Fig. S11, in which the transported BC and local BC profiles resemble the No_EMISS_NE and Only_EMISS_NE PM$_{10}$ profiles, respectively. IGP was the dominant source of transported BC (Fig. S12). In transported BC scenarios, BC was available up to much higher atmospheric height and profiles showed elevated concentration at around 1500 m indicating stronger BC transport at that height. In Only_NE_BC and Only_NE_2×BC, BC was confined near the surface, which decreased continuously. The atmospheric heating rate (HR) was estimated as per Liou (1980).

$$HR = \frac{g}{C_p} \cdot \frac{\Delta F}{\Delta P}, \tag{4}$$

where g is the acceleration due to gravity (9.81 m s$^{-2}$), $C_p$ is the specific heat capacity of air at constant pressure (1.005 kJ K$^{-1}$ kg$^{-1}$), $\Delta F$ the atmospheric RF and $\Delta P$ is the atmospheric pressure (300 hPa) difference between surface and 3 km altitude as most of the BC was present below this height. Moreover, in order to compare the effectiveness of heating by local and transported BC, two parameters, heating efficiency (HE) and heating slope (HS), were defined by equations 5 and 6.

$$HE = \frac{HR}{\text{Column sum of BC concentration within 3 km (CC)}}, \tag{5}$$

$$HS = \frac{\Delta HR}{\Delta CC}, \tag{6}$$

HE has units of K day$^{-1}$ µg$^{-1}$ m$^3$, thus measuring the heating contributed by per unit concentration of BC below 3 km. HE was used to assess the effect of BC vertical distribution on atmospheric heating while HS was used to assess the response of atmospheric heating rate to BC concentration changes and has similar units as HE. CC has units of µg m$^{-3}$.

Table 4: NE India region average values of columnar BC concentration (µg m$^{-3}$) and atmospheric heating
parameters in different scenarios

| | No_NE_BC | No_NE_2×BC | Only_NE_BC | Only_NE_2×BC |
|---|---|---|---|---|
| **HR** | 0.460 | 0.597 | 0.123 | 0.178 |
| **CC** | 12.458 | 18.391 | 3.905 | 7.563 |
| **HE** | 0.037 | 0.032 | 0.032 | 0.024 |
| **ΔHE** | -0.004 | | -0.008 | |
| **HS** | 0.023 | | 0.015 | |

The quantitative values of the parameters are provided in Table 4. Only_NE_BC had a regional net average HR
of 0.123 K day$^{-1}$ compared to 0.460 K day$^{-1}$ of No_NE_BC. This indicated a 3.73 times higher atmospheric heating
rate by transported BC. An increase in local emissions from Only_NE_BC to Only_NE_2×BC caused a small
increase in heating rate of 0.055 K day$^{-1}$ compared to the increase of 0.137 K day$^{-1}$ from No_NE_BC to
No_NE_2×BC. As per the definition, HE was inversely proportional to CC and this was exactly followed in all
regions across all scenarios (Fig. S13 and S14). However, HE was higher in the case of transported BC compared
to local BC with values of 0.037 K day$^{-1}$ µg$^{-1}$ m$^3$ (No_NE_BC) vs. 0.032 K day$^{-1}$ µg$^{-1}$ m$^3$ (Only_NE_BC) and
0.032 K day$^{-1}$ µg$^{-1}$ m$^3$ (No_NE_2×BC) vs. 0.024 K day$^{-1}$ µg$^{-1}$ m$^3$ (Only_NE_2×BC), even if CC was higher in the
case of transported BC. The reason might be that transported BC might have undergone a higher amount of
chemical transformation due to higher atmospheric time, leading to a higher lensing effect on the BC core,
resulting in enhanced absorption (Liu et al., 2015). Also, it was observed that on increasing emissions, the decrease
in HE was smaller in the case of transported BC (-0.004 K day$^{-1}$ µg$^{-1}$ m$^3$) than local BC (-0.008 K day$^{-1}$ µg$^{-1}$ m$^3$).
Hence, with the increase in BC emissions, HE decreased more when BC was more concentrated near the surface
than in the atmosphere. HS indicated that atmospheric heating increased at a higher rate of 0.023 K day$^{-1}$ µg$^{-1}$ m$^3$
with increasing transported BC compared to 0.015 K day$^{-1}$ µg$^{-1}$ m$^3$. Thus, the increase in transported BC emissions
had more impact on atmospheric heating over NE India than when present near the surface with local emissions.
**3.4.2 Atmospheric stability and moisture**

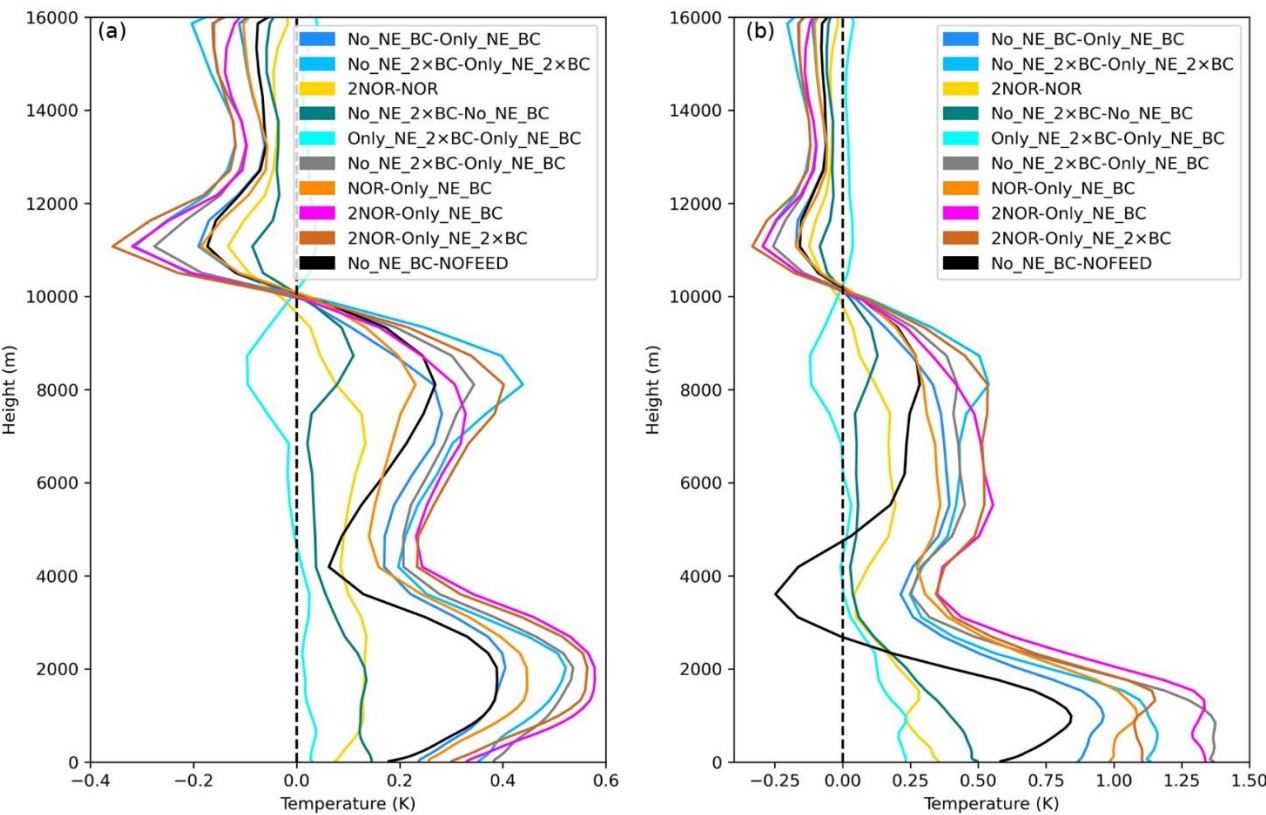


Figure 7: Regionally averaged vertical profiles showing perturbations in a) potential temperature (K) b) equivalent potential temperature (K)

Barman and Gokhale (2022), as well as Soni et al. (2017), showed an increased influx of moisture into the region during pre-monsoon due to BC. In order to compare and separate the effects of local and transported BC on atmospheric stability through temperature and moisture, potential temperature (PT) and EPT were estimated. PT estimates atmospheric stability based on temperature, while EPT accounts for both temperature and moisture and is a more realistic parameter. In most of the profiles in both parameters in Fig. 7(a) and 7(b), positive perturbation was observed approximately below 10 km and negative above it which indicated an increase in atmospheric instability and vice-versa for an increase in atmospheric stability(Zhao et al., 2011). The positive perturbations below 10 km varied with height and were most profound in the profiles No_NE_BC – Only_NE_BC, No_NE_2×BC – Only_NE_2×BC and No_NE_2×BC – Only_NE_BC, each of which was estimated from the difference between a transported BC scenario and local BC scenario. These profiles showed similarity with the corresponding profiles of NOR – Only_NE_BC, 2NOR – Only_NE_2×BC and 2NOR – Only_NE_BC in both the parameters, indicating that they were closer to the normal atmospheric scenario. The positive perturbations were, however, comparatively smaller with 2NOR – NOR, No_NE_2×BC – No_NE_BC and Only_NE_2×BC – Only_NE_BC in both the parameters, each pair being the same scenario with only a difference in emission rates. This shows that BC atmospheric distribution played an important role on instability. The Only_NE_2×BC – Only_NE_BC profile not only showed a smaller increase in instability than No_NE_2×BC – No_NE_BC profile but also contributed to the smallest increase in instability in both the parameters. Thus, transported BC and an increase in transported BC emissions led to higher atmospheric instability than local BC.

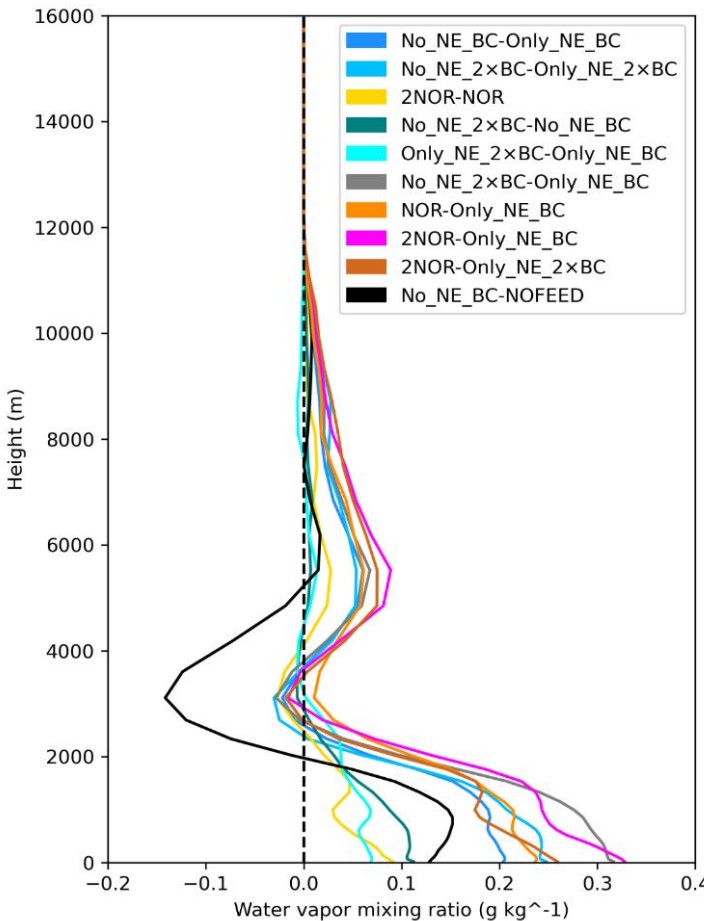

386

Figure 8: Regionally averaged vertical profiles showing perturbations in water vapor mixing ratio (g kg$^{-1}$)

Moreover, EPT profiles showed higher positive perturbations and hence higher instability compared to the corresponding PT profiles with values exceeding 1.25 K. The positive difference or additional instability between the corresponding profiles of Fig. 7(a) and 7(b) was due to moisture. The difference also indicated that moisture contributed even more to the instability than BC. The peaks for EPT existed closer to the surface due to most of the moisture also remaining near the surface, as shown in Fig. 8. However, there occurred a region of increased stability from the ground surface to the first peak of transported BC profiles at approximately 1000 m, indicated by increasing temperature with height. Thus, transported BC may also be responsible for air quality scenarios over NE India by creating a stable boundary layer. The close qualitative and quantitative similarity between No_NE_BC – NOFEED, No_NE_BC – Only_NE_BC and NOR – Only_NE_BC profiles in Fig. 7(a) showed that aerosol radiative effect due to transported BC was intricately linked with the PT profile and the positive perturbations in each of these profiles were also closely linked with BC. This was also seen in Fig. 7(b), but since it also included the effect of moisture, larger differences were seen.

BC, whether transported or emitted locally, caused a positive perturbation in moisture at least below 2 km altitude, as seen in Fig. 8. The perturbation was much larger in profiles that had a combination of transported and local BC scenarios and which had higher transported BC emissions and followed the pattern similar to PT and EPT. This links BC, instability and moisture in the region, i.e., higher transported BC caused higher instability which brought a higher amount of moisture which would possibly again cause higher instability. It was the same

for scenarios that included indirect effect, as can be observed from the similarities of the No_NE_BCI –
Only_NE_BCI (Fig. S15) and No_NE_BC – Only_NE_BC profile in Fig.8. Furthermore, the similarity of
No_EMISS_NE – Only_EMISS_NE profile with No_NE_BCI – Only_NE_BCI (Fig. S15) inferred that direct
radiative effect of transported BC was responsible for the moisture increase in Fig. 4. The higher moisture with
transported BC scenarios was due to higher moisture flux caused by it over Bay of Bengal compared to local BC
and can be verified from Fig. 9. Quantitatively, No_NE_BC (33.95 kg m$^{-2}$) and No_NE_2×BC (34.15 kg m$^{-2}$) had
higher region average precipitable water vapor than Only_NE_BC (33.49 kg m$^{-2}$) and Only_NE_2×BC (33.64 kg
m$^{-2}$). Hence transported BC in Sect. 3.3 was primarily responsible for transporting moisture from the Bay of
Bengal by affecting the atmospheric dynamics. The mechanism is similar to the "heat pump" model by Lau et al.
(2006).

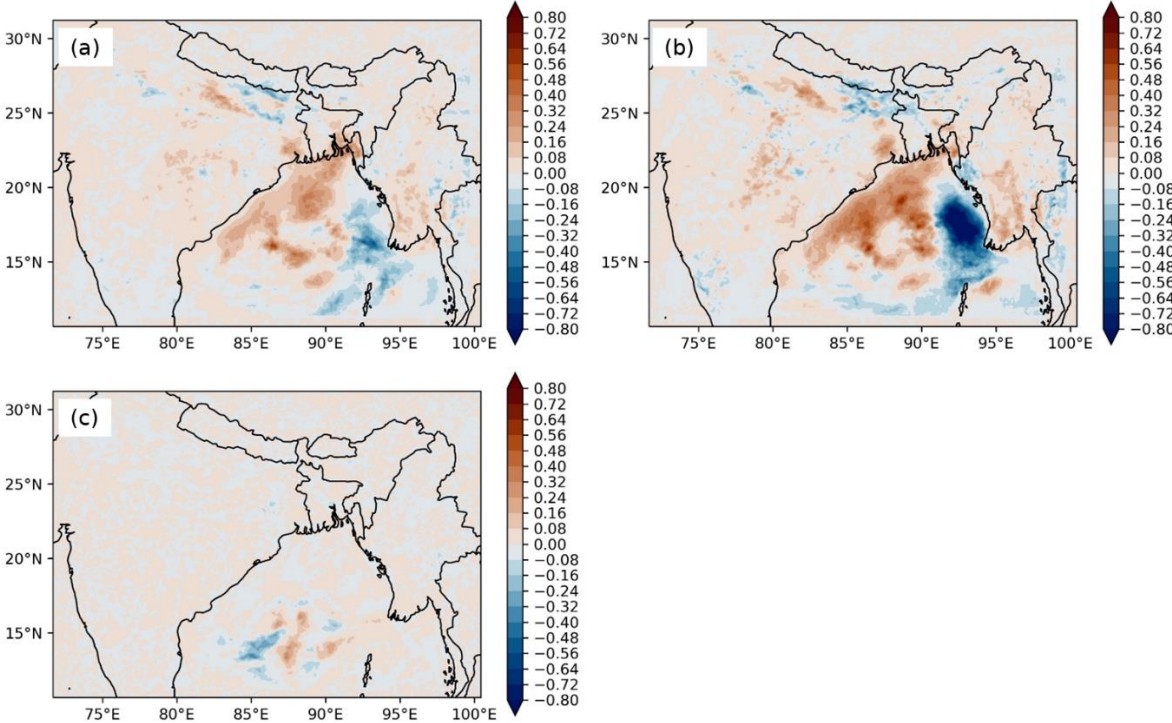

Figure 9: Spatial distributions of change in moisture flux (g s$^{-1}$ m$^{-2}$) in a) No_NE_BC – Only_NE_BC b)
No_NE_2×BC – Only_NE_BC c) Only_NE_2×BC – Only_NE_BC near surface

**3.5 Rainfall response to emissions**

Similar to NOR-I – NOCHEM, No_BC_ABS – NOCHEM gave the rainfall change due to total aerosol effect,
but without BC absorption. The higher negative rainfall change of -275.13 mm with NOR-I – NOCHEM
compared to -266.78 mm with No_BC_ABS – NOCHEM showed BC absorption to reduce rainfall. The higher
reduction with NOR-I – NOCHEM was mainly due to higher rainfall reduction in region 1, where the direct and
semi-direct effect was maximum. This shows BC initially suppressed rainfall even though moisture increased due
to it. However, with the increase in BC emissions, rainfall increased and the rainfall suppression due to the total
aerosol effect reduced substantially to -64.44 mm with 4NOR-I – NOCHEM compared to -275.13 mm with NOR-
I – NOCHEM and similarly, rainfall due to direct and semi-direct with 4NOR-I – NOFEED-I showed a positive
rainfall change of 193.64 mm compared to -17.04 mm with NOR-I – NOFEED. Similarly, 4NOR-I – NOR-I gave

a rainfall enhancement of 225.24 mm. Spatial distribution of change in rainfall is shown in Fig. 10(a) which show
rainfall change primarily occurring over NE India and along the valley.

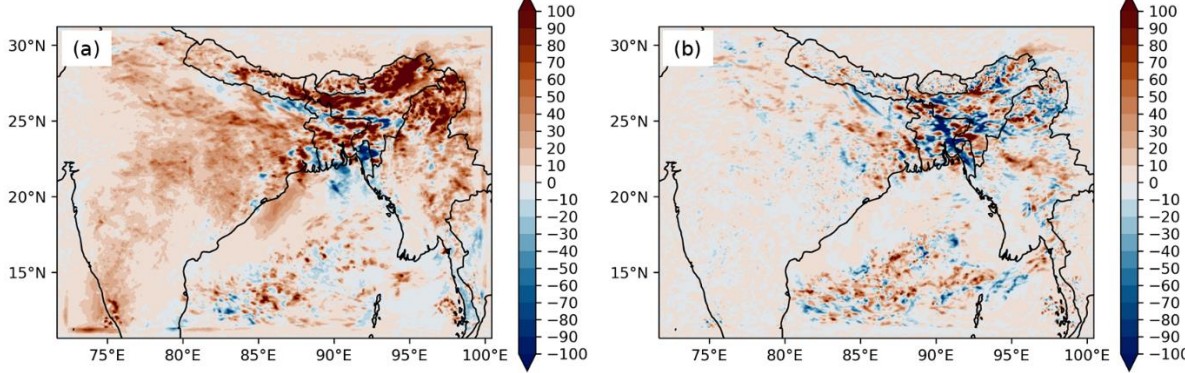


Figure 10: Spatial distributions of change in rainfall (mm) in a) 4NOR-I – NOR-I b) No_EMISS_NE_4SO$_2$ –
No_EMISS_NE_0.25SO$_2$
Aged BC also contributes as CCN (Lambe et al., 2015). The enhancement in BC emission did increase the column
average CCN concentration to 2252 m$^{-3}$ (4NOR-I) from 2024 m$^{-3}$ (NOR-I), but the increase was largely
disproportionate to the 4 times BC emission increase. The enhancement over NE India can also be seen from the
spatial distribution of column integrated CCN in Fig. 11.

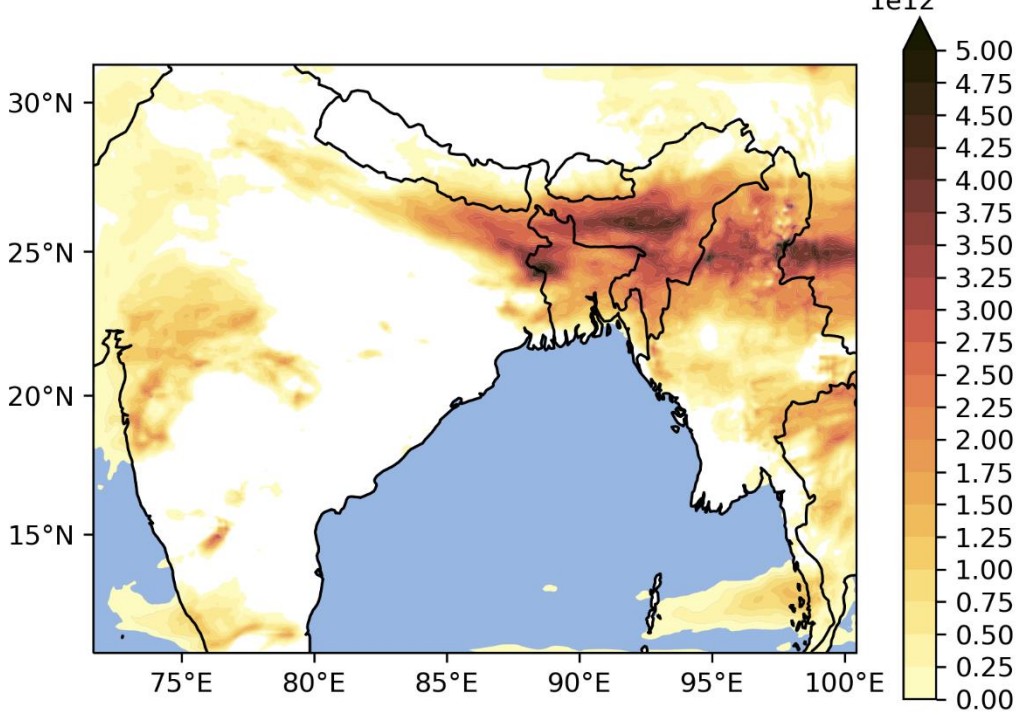


Figure 11: Spatial distribution of column integrated CCN number (m$^{-2}$), estimated from 4NOR-I – NOR-I
Enhancement of CCN number concentration generally leads to enhancement of indirect aerosol effect (Yu et al.,
2013) and also seen later in case of sulfate aerosol. However, in spite of the increase in CCN, cloudwater mixing
ratio was lower in 4NOR-I than NOR-I, as seen in Fig. 6 and 4NOR-I caused significantly more rainfall formation
than NOR-I, as can be seen from the rainwater mixing ratio profiles. This may be related to the suppression of
CCN activation due to BC, as observed over Central India (Nair Jayachandran et al., 2020). Also BC contributes
marginally to indirect effect (Kristjánsson, 2002). Thus, the increased moisture (Fig. S15) did not remain stored
as cloudwater even though there was an increase in CCN, but it got converted to rainwater. The large increase in
moisture, caused by the increase in atmospheric instability possibly condensed on relatively a smaller number of
CCN particles promoting larger cloud droplets which enhanced rainfall. Moreover, the ratio of rainwater mixing
ratio to rain droplet number concentration gave the amount of rain water per rain droplet, or indirectly the rain
droplet size. The vertical profile of this ratio is shown in Figure 12, which shows higher values for 4NOR-I.

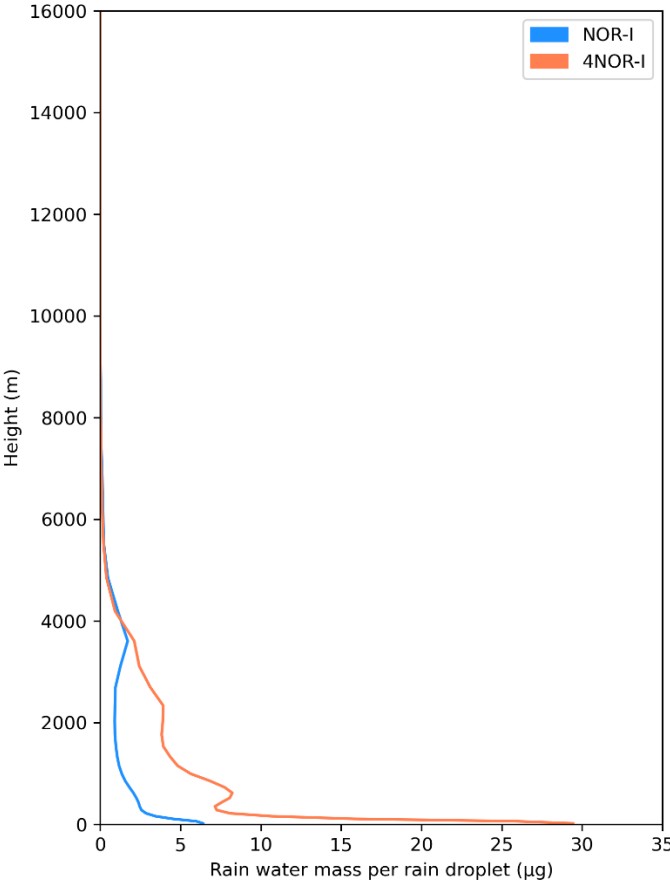

Figure 12: NE India region averaged vertical profiles of rain water mass per rain droplet
Collision is the primary mechanism of rain development in warm clouds (Lamb and Verlinde, 2011). Since rain
droplets are formed from the gathering of cloud droplets, the higher value for 4NOR-I indicated larger rain droplet
formation, possibly through better collisions among the cloud droplets, besides higher moisture availability. This
indicated that the increase in BC emissions didn't contribute to rainfall suppression through indirect aerosol effect
though there was an increase in CCN concentration, but rather counteracted the suppression of rainfall due to the
indirect effect of other aerosol species. The rainfall enhancement was due to an increase in moisture, contributed
by the transported fraction of BC, as explained in Sect. 3.4.2. Moreover, rainfall suppression was also more due
to transported aerosols, mainly contributed by indirect effect (Table 3). Also, among the non-absorbing aerosols,
sulfate aerosol is an important contributor to CCN and indirect effect (Kristjánsson, 2002). Its concentration was
found to be the highest among non-absorbing aerosols and most of its mass over NE India was found to be

transported (Sect. 3.1). Concentration profiles can be seen from Fig. S16. Hence, the response of rainfall over NE India was checked by increasing (No_EMISS_NE_4SO$_2$) and decreasing (No_EMISS_NE_0.25SO$_2$) SO$_2$ emissions outside NE India and compared against the baseline transported scenario (No_EMISS_NE) since sulfate is mainly formed within the atmosphere by oxidation of SO$_2$ (Wang et al., 2021). Similar to the increase in BC emissions, No_EMISS_NE_4SO$_2$ caused an increase in column average CCN concentration to 3524 m$^{-3}$ compared to 1753 m$^{-3}$ in No_EMISS_NE, while No_EMISS_NE_0.25SO$_2$ showed a decrease (1390 m$^{-3}$). However, contrary to the BC, an increase in SO$_2$ emissions with No_EMISS_NE_4SO$_2$ caused an increase in cloudwater mixing ratio compared to No_EMISS_NE, as seen in Fig. 6, while its decrease also caused a decrease. Thus, No_EMISS_NE_4SO$_2$ and No_EMISS_NE_0.25SO$_2$ had lower and higher rainwater mixing ratio, respectively, compared to No_EMISS_NE. Consequently, No_EMISS_NE_4SO$_2$ had higher rainfall suppression and gave lesser rainfall (-22.23 mm) compared to No_EMISS_NE_0.25SO$_2$. Spatial distribution is shown in Fig. 10(b) which show mainly negative change over the region. Thus, an increase in non-absorbing aerosol caused rainfall suppression through indirect effect. The indirect effect was observed to be the dominant aerosol effect for suppressing rainfall. However, with an increase in BC, suppression of rainfall due to direct and semi-direct effects through surface processes (surface moisture flux, convection) and cloud evaporation as well as due to indirect aerosol effect (atmospheric stability, surface moisture flux and cloud to rainwater conversion) becomes comparatively weaker mechanisms than the direct effect of radiative heating by BC, enhancing rainfall through the transport of moisture. However, the increase in transported SO$_2$ emissions also caused further suppression of rainfall. Hence, an increase in transported aerosols of an absorbing aerosol (BC) and a non-absorbing aerosol (sulfate), both being a contributor to CCN, exerted different impacts to indirect effect parameters and thus to rainfall and hence most likely controls the enhancement and suppression of pre-monsoon rainfall over NE India, thus counteracting each other. However, since decreasing rainfall trend has been observed, the impacts of the indirect aerosol effect could be dominant. Here, the response of only one non-absorbing aerosol (sulfate) was checked and possibly has contributions from other similar species also. Other non-absorbing aerosol species like nitrate also contribute to indirect aerosol effect (Wang et al., 2010; Zaveri et al., 2021) which may contribute to rainfall suppression as sulfate.

Moreover, the percentage of the simulation time different aerosol effects and BC emissions increased (inc) or suppressed (dec) rainfall under different rainfall intensities (low: 0-5, medium: 5-10, high: > 10 mm day$^{-1}$; defined as per (Raju et al., 2015)) and the rainfall amount under those intensities was estimated. Regional average values are provided in Table S6 and S7. All aerosol effects caused a higher decrease across all rainfall intensities, except the indirect effect, which indicated a higher increase in low-intensity rainfall (6.52 mm vs. -6.48 mm; 21.44 % vs. 20.58 %). High-intensity rain was primarily responsible for rainfall changes across all the scenarios and effects. The indirect effect decreased high-intensity rainfall duration (18.85 vs. 12.38 %) and amount (-399.41 mm vs. 141.62 mm) and was primarily responsible for the rainfall suppression in total aerosol effect (-411.34 mm). The total aerosol effect with enhanced BC emissions (4NOR-I – NOCHEM) showed a significantly higher increase (275.47 mm vs. 137.16 mm) as well as a significantly lower decrease (-337.23 vs. -411.34) in high-intensity rainfall compared to total aerosol effect with baseline BC emissions (NOR-I – NOCHEM). Similar results in time and rainfall amount between BC increase and direct + semi-direct effect with BC increase scenarios inferred that enhanced radiative effects due to BC increase were mainly responsible for higher high-intensity rainfall duration and rainfall amount, while the indirect aerosol effect was mainly involved in its suppression,

possibly due to the increased atmospheric stability associated with it. Barman and Gokhale (2022) also showed
similar results with BC emissions increase, but this study verifies the role of direct radiative effects of BC in it.
Thus, BC increased rainfall over NE India but in the form of high-intensity rainfall. Hence, relative fractions of
BC and the other aerosols contributing to indirect effect possibly decide the amount of rainfall and its intensity
over the region. However, indirect effect also caused high-intensity rainfall but with lesser amount than its
suppression and may be involved in catastrophic flood events at local scales (Wang et al., 2022).

**4 Conclusions**

Transported aerosols, primarily from IGP, were found to be responsible for the bulk of the aerosol mass (93.98
%) over NE India while contributing 64.18 % of near-surface $PM_{10}$ concentration, thus primarily responsible for
air pollution as climatic impacts over the region during pre-monsoon season. The climatic impacts, both w.r.t. RF
as well as rainfall, were dominated by the indirect aerosol effect. The impacts of the indirect aerosol effects of
transported aerosols were much higher in affecting radiation (-13.12 W $m^{-2}$ vs. -0.24 W $m^{-2}$ at the surface, 7.30 W
$m^{-2}$ vs. 0.97 W $m^{-2}$ in the atmosphere) as well as suppressing rainfall (-49.11 mm vs. -16.04 mm) compared to
local emissions. The greater surface dimming by transported aerosols caused a higher negative change in surface
moisture flux (-3.82×$10^{-6}$ kg $m^{-2}$ $s^{-1}$ vs. 8.15×$10^{-8}$ kg $m^{-2}$ $s^{-1}$) as well as higher aerosol mass reduced cloudwater to
rainwater conversion, both of which contributed to higher rainfall suppression. Transported aerosols caused
4.42×$10^{13}$ $m^{-2}$ higher cloud droplets than local emissions. The atmospheric instability due to the direct + semi-
direct effect and indirect effect of transported aerosols were found to be contradictory and caused an increase and
decrease, respectively. The direct effect of transported aerosols, though also caused negative surface moisture flux
over NE India (-1.03×$10^{-6}$ kg $m^{-2}$ $s^{-1}$), however, increased moisture over NE India, increasing moisture flux over
the Bay of Bengal. Further analysis showed that transported BC was more efficient in atmospheric heating over
NE India and together with the higher transported BC mass, an increase in its emissions caused higher atmospheric
instability over the region, which brought more moisture from the Bay of Bengal. The increased moisture further
contributed to higher instability. Hence, the rainfall suppression caused through the different atmospheric
processes by direct, semi-direct and indirect effects was reduced and nullified with the increase in BC emissions,
but the rainfall increase was mainly in the form of high-intensity rainfall. The increase in BC did not show a
positive change in cloudwater, though it contributed to CCN. The direct effect of BC thus overpowered the other
rainfall-suppressing processes. Indirect aerosol effect and radiative heating were the main rainfall-controlling
factors. Hence, changes in emissions of aerosols or chemical species contributing to these processes will possibly
contribute to rainfall suppression and enhancement over NE India. Moreover, rainfall simulated with transported
aerosols were found to be more similar to the IMD observation datasets as well as the baseline emission scenario,
indicating its possible greater influence in the real-world scenario.
The study shows that the atmospheric transport of emissions from IGP to NE India has a significant
impact on NE India's rainfall during pre-monsoon and the impacts are even greater than the emissions within the
NE India region.
**Data availability.** Model outputs are available upon request.
**Author contributions.** NB - conceptualization, methodology, model simulation, visualisation, manuscript
writing, SG - conceptualization, methodology and supervision, manuscript review and editing.
**Competing interests.** The authors declare that they have no conflict of interest.
**Disclaimer.** The views expressed in this paper are those of the authors.
**Acknowledgements.** The simulations were performed on the "Param-Ishan" HPC of Indian Institute of
Technology Guwahati. The authors are also grateful to "Air and Noise Pollution Lab" of Civil Department IIT
Guwahati for their support.

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
