# Peer review of "Transported aerosols regulate the pre-monsoon atmosphere over North-East India: a"

_EGUsphere, 2023_

## Author Response (AR1)

**Reviewer comments 1**

1. The "pre-monsoon atmosphere" in the title seems not specific and less informative. What does the "atmosphere" refer to? Temperature, humidity, stability, precipitation, or chemical components? According to the abstract and analysis in the paper, I assume the author intended to focus on the rainfall and explain its decreasing trend. In that case, you may consider narrowing down the title to "the decreasing rainfall".

Author' s response: Rainfall was the primary objective of the study but as was observed from the results, the chemical components affected several components of the atmosphere starting from solar radiation, moisture availability, atmospheric instability and clouds which ultimately influenced rainfall. Hence to underline the importance of effects on a large number of components of the atmosphere as a whole, the word "atmosphere" was used.

Changes in manuscript: However, as per suggestion, the title has been changed to "Transported aerosols regulate the pre-monsoon rainfall over North-East India: a WRF-Chem modelling study" and hence "atmosphere" in line 9 and 535 also replaced by "rainfall".

2. The above comment leads to my biggest concern of this study. The author stressed the aerosols effect on rainfall so much yet no figure regarding the rainfall is shown in the main text. The water vapor mixing ratio only indicates the availability or probability of precipitation, and the actual response of rainfall due to transported absorbing and scattering aerosols should be displayed. Is the aerosol indirect effect similar to or different from that over IGP in suppressing rainfall and why? Is there any evidence of water vapor transport from Bay of Bengal? The author may need more figures in the main text to elucidate the point.

Author' s response: A previous study (https://doi.org/10.1007/s00704-017-2057-1), referred in Section 3.4.2 also showed BC to increase moisture transport from Bay of Bengal into North East India and increase rainfall over this region. However, this study adds that the BC outside NE India is mainly responsible for it.

Water vapor is the primary ingredient for rainfall but its conversion to rainfall is influenced by the aerosols, acting as cloud condensation nuclei (CCN). The presence of large amount of CCN facilitates condensation of water vapor on numerous aerosol particles, producing numerous cloud droplets with smaller radii. This restricts small cloud droplets to grow in size due to reduction in interaction with other cloud droplets which affect its conversion to rain droplet and thus to rainfall. This is termed as aerosol indirect effect. However, BC was found to be inefficient as CCN when its emission was increased from NOR-I to

4NOR-I. Hence the large increase in moisture, caused by the increase in atmospheric instability possibly condensed on relatively a smaller number of CCN particles promoting larger cloud droplets which enhanced rainfall. Rainfall suppression due indirect aerosol effect is defined similarly irrespective of location and hence also same for IGP. The mechanism is as explained above.

Changes in manuscript:

1. Description regarding rainfall enhancement by BC, as explained in above paragraph has been added to the main text in Section 3.3 at lines 288-292 and in Section 3.5 at lines 444-447 for better clarity.

2. Figures showing rainfall response calculated from 4NOR-I – NOR-I and No_EMISS_NE_4SO$_2$ - No_EMISS_NE_0.25SO$_2$ has been added as Figure 10. 4NOR-I – NOR-I is not exactly the transported scenario but as seen from the results the local BC component was ineffective in affecting moisture or atmospheric stability and hence majorly represents the effect of transported BC component in it.

3. Figure S10 has been moved to main text as Figure 9 which shows increased water vapor transport due to transported BC.

3. As a research greatly based on numerical simulations, the illustration of the model experiments is much deficient with many important details missing. For example, what are the physical and chemical parameterizations of WRF-Chem? How did the author configure the model domain and which area is referred to as NE India? How did the author determine the anthropogenic and natural (i.e., dust and biogenic) emissions? When it comes to experiment with emissions only within NE India does it mean all emissions or mainly anthropogenic emissions? In addition, the evaluation of model performance against MERRA-2 is less convincing since MERRA-2 is also generated by global atmospheric model with assimilation system. In-situ observations concerning chemical components of the atmosphere should be involved in model verification.

Author' s response: The model domain is shown in Fig. 1(a) and the NE India is the part of India within the region bounded by the blue box. The region within the box is bounded by 22° N and 29° N latitudes and 89° E and 97° E longitudes. During the study period the near surface wind flow was from the Bay of Bengal towards NE India, which gradually changed to westerly wind flow carrying aerosols from IGP towards NE India. Hence the domain was selected by keeping the NE India region near the upper-right corner of the domain. As per suggestion, comparison against in-situ observation stations were also carried out.

Changes in manuscript:

1. Physical and chemical parametrizations have been added in Section 2 in Table 1.

2. Better description of simulations has been added in Section 2 for NOR, NOR-I and some other simulation experiments in lines 88-101. Remaining simulations can be understood from Table 2 by it comparing with it.

3. The NE India region description has been added in Section 2 in lines 95-96.

4. Description regarding selection of the domain has been added in Section 2 in lines 83-84.

5. Details regarding emissions and model inputs have been added in Section 2 in Table 1.

6. All emissions (natural and anthropogenic) were considered with emissions only within NE India and mentioned in Section 2 in line 95.

7. Comparison against in-situ observations have been included in Section 2 in lines 125-132 and performance statistics included in Table S3.

Minors:

4. Line 11: change "and" to "on".

Author' s response: Correction done.

Changes in manuscript: Changed "and" to "on".

5. Line 17: "the aerosols effects were observed to be ..." this sentence may cause ambiguity since the aerosols effects in this study are all analyzed by models.

Author' s response: Correction done.

Changes in manuscript: Removed "observed to be".

6. Line 119: what do you mean by "atmospheric distribution"?

Author' s response: It meant the three-dimensional distribution.

Changes in manuscript: For better understanding, it has been replaced by "vertical distribution".

7. Line 144: The BC and sulfate aerosols are all components of $PM_5$. Why does the author focused on $PM_{10}$ instead of $PM_{2.5}$? How about the spatial distribution of $PM_{2.5}$?

Author's response: This study is mainly a climatic study where aerosols of all size ranges have climatic impacts. Also, the maximum size supported in the model is 10 μm. Hence $PM_{10}$ mass was reported in order to later relate with the impacts. The spatial and vertical distribution of $PM_{2.5}$ was found to be more or less similar to the $PM_{10}$ with NOR-I, No_EMISS_NE and Only_EMISS_NE, albeit with lower concentration values.

Changes in manuscript: No changes.

8. Figure 1: what do rectangles in panel(a) and (d) mean?

Author's response:

1. Rectangle in Figure 1(a) includes the North-East region of India. See response #3 to comment #3.
2. The North-East India region was divided into 4 sub-regions, which is shown in Figure 1(d).

Changes in manuscript: Better description regarding the 4 sub-regions has been added in Section 3.1in lines 144-147. See response to comment #10.

9. Line 123: what's the height of level 0 and level 15 of your model? Please be specific.

Author's response: The height of the model levels vary spatially according to the height of the terrain as the model levels are terrain-following. Hence any model level does not have any specific height. The height of the model level is the geopotential height and the spatial distribution of it are provided in the supplementary Figure S1. The height of concentration values for Figure 1(a), 1(b) and 1(c) is Figure S1(a) and similarly the height of concentration values for Figure 1(d), 1(e) and 1(f) is Figure S1(b).

Changes in manuscript: No changes.

10. Line 145: why is NE India separated into four sub-regions? Please explain.

Author's response: The NE India region was divided into four sub-regions based on the proximity from the IGP. Region 1 and region 2 fall along the Brahmaputra River valley, but with region 1 being closest to IGP. Region 3 is mostly a mountainous region. Region 4 is the southern region closer to the Bay of Bengal.

Changes in manuscript: Better description regarding it has been added in Section 3.1 in lines 144-147.

11. Line 161: "… and However …" is not a complete sentence.

Author' s response: Agreed.

Changes in manuscript: removed "and"

12. Figure 4 and 5: the title of X-axis is missing in these two plots.

Author' s response: Agreed.

Changes in manuscript:  Titles have been added.

13. Line 285: it is confusing why the author still discussed the role of locally emitted BC since chapter 3.3 just clarified the role of local aerosols is not important at all.

Author' s response:  The analysis with the locally emitted BC provided a further verification of the results in Section 3.3. Moreover, the analysis in Section 3.4 provides a comparison of the impacts of local and transported BC which showed that increase in locally emitted BC emissions has much lesser impact than transported BC and hence more focus can be given on the regions responsible for the transported BC, i.e. IGP.

Changes in manuscript:  No changes.

14. Line 425: the transported aerosols "exert different impacts" on cloud parameters and rainfall, not "show responses to".

Author' s response: Agreed.

Changes in manuscript:  Changed "show responses to" to "exerted different impacts"

**Reviewer comments 2**

The study showed the transported aerosols from Indo-Gangetic Plain (IGP) in northern India to have significant effect on meteorology, i.e. suppressing rainfall over North-East (NE) India, utilizing several numerical experiments with the WRF-Chem model. The work is quite detailed and generally well-documented with suitable citations across the manuscript text. The main finding of this paper highlights significance of IGP as a hotspot of diverse aerosols, particularly black carbon (BC), which has a significant impact on suppressing rainfall in NE India via the indirect aerosol effect.

**Main comments:**

1. However, my main comment is on that aspect, Figure S10 which I believe authors shifted as per suggestion of Reviewer 1 to main text does highlight that the impacts of transported BC from IGP are greater than the emissions within the NE India region. To support the main argument, it would be better to include a Lagrangian Back-Trajectory (using model such as HYSPLIT: https://www.ready.noaa.gov/HYSPLIT.php) for pre-monsoon (main inference of this study) to show the transport of air-mass from IGP to NE India in this season (around the April 2018 period considered in modeling), if possible. Or alternatively elaborate more and/or provide suitable citations that have shown such transport earlier to support statements such as, in Lines 37-38: "The condition becomes more critical in the pre-monsoon season when the westerlies directly transport air pollutants from the IGP to NE India."

   Author' s response: HYSPLIT analysis has not been done but streamlines show the air-mass flow from IGP to NE India.

   Changes in manuscript: Using the model output, streamlines showing the air-mass flow from IGP to NE India along with the $PM_{10}$ mass flux has been added as Figure 2.

2. Lines 393-399: "However, in spite of the increase in CCN, cloudwater mixing ratio was lower in 4NOR-I than NOR-I, as seen in Fig. 5 and 4NOR-I caused significantly more rainfall formation than NOR-I, as can be seen from the rainwater mixing ratio profiles. This may be related to the suppression of CCN activation due to BC, as observed over Central India (Nair Jayachandran et al., 2020). Also BC contributes marginally to indirect effect (Kristjánsson, 2002). Thus, the increased moisture (Fig. S9) did not remain stored as cloudwater even though there was an increase in CCN, but it got converted to rainwater." Some discussion on cloud condensation nuclei

(CCN) number concentrations specially focusing on NE sub-domain(s) is warranted as well, along with addition of the geospatial patterns of CCN (at least for relevant experiments, if not all). There is some discussion on CCN number concentration in Section 3.5 (Rainfall response to emissions) but complementing it with a figure is necessary to make the authors argument more robust. This adds to the Reviewer 1's comment on need of some key figures (assuming the figure for rainfall is already added in the next revision) in the main text to elucidate the necessary argument(s).

Author' s response: Spatial distribution shows a positive change in CCN number but a positive rainfall change. Moreover, the rainwater mixing ratio/rain droplet number ratio indicating rain droplet size inferred the formation of larger rain droplets in 4NOR-I compared to NOR-I. This inferred better collisions among the cloud droplets in 4NOR-I.

Changes in manuscript: The spatial distribution of change in column integrated CCN, estimated from 4NOR-I – NOR-I has been added in Figure 11, which also show increase in CCN over NE India. Moreover, vertical profile of rain water mass per rain droplet has been added in Figure 12 and discussion added in lines 447-454.

**Minor comments:**

1. Boundaries with LAT-LON of the model domains (please mention them in the figure caption as well) used in this study, should be added to a current figure at the beginning of the manuscript.

   Author' s response: Changes accepted.

   Changes in manuscript: Latitude and longitude of the boundaries of the model domain has been added to the figure at the beginning of the manuscript. However, the figure is a graphical abstract for which no caption is to be added as per the format of the ACP and but has been specified in the main text at line 81.

2. Physical and chemical parameterizations along with source of varied emissions used in the WRF-Chem study should be summarized as a table.

   Author' s response: Changes accepted.

   Changes in manuscript: Physical and chemical parameterizations as well as emissions has been added as Table 1.

3. Table 2: Correct the formatting to not be in-line with the text (currently line numbers are getting superimposed over the table making it unclear).

Author' s response: Changes accepted.

Changes in manuscript: Correction has been done.

4. Line 73: Correct to "outputs from 10-19 April 2018 were used for analysis."

Author' s response: Changes accepted.

Changes in manuscript: Correction has been done.

5. Lines 124-125: "The spatial distribution of geopotential heights of model level 0 and 15 are shown in Fig. S1…" While it is understandable that geospatial heights of model layer would vary across the whole wider Indian subcontinent domain shown in Figure S1. But would be better to provide a range for them, say for IGP and NE domain regions. This is also critical in terms of supporting inferences such as in : Lines 328-329: "Thus, the increase in transported BC emissions had more impact on atmospheric heating over NE India than when present near the surface with local emissions."

Author' s response: Changes accepted.

Changes in manuscript: Separate ranges have been provided for model level 0 and 15 in Fig. S1.

6. Lines 144-145: "Further analysis indicated that transported aerosols accounted for >50 % of BC, organic carbon, sulfate, nitrate, ammonium and dust aerosol mass over NE India's atmosphere." Please support this finding with additional figures (Also see main comment#1).

Author' s response: Changes accepted.

Changes in manuscript: The column integrated mass of these species has been added in lines 173-176, which supports the given statement and also Figure S3-S8 has been added.

7. Lines 165-166: Correct to "…as it contributed to greater cloud cover (Nandan et al., 2022), which caused heating of the surface through LW radiation." Please correct any similar grammatical issues before submitting the revised version.

Author's response: Changes accepted.

Changes in manuscript: Correction has been done.

8. Please ensure that plot axes and panels are labeled, in figures, wherever appropriate for clarity.

Author's response: The axes of all figures, except those showing spatial distributions have been labelled. The caption of the figures also mentions the units and provide adequate clarity.

Changes in manuscript: Unit "(K)" has been added to caption of Figure 7.

9. Lines 428-429: "Here, the response of only one non-absorbing aerosol (sulfate) was checked and possibly has contributions from other similar species also." Please elaborate more on the possible impacts other species may have with suitable citations included.

Author's response: Other non-absorbing species such as nitrate also contribute to indirect aerosol effect and may have similar rainfall suppressing effect as sulfate.

Changes in manuscript: Suitable citations have been added in lines 486-487.

10. Lines 469-471: "The increase in BC did not show a positive change in cloudwater, though it contributed to CCN. The direct effect of BC thus overpowered the other rainfall-suppressing processes." (Refer to main comment #2, additional discussion with figure(s) needed to be tied with such inferences in discussion of results).

Author's response: Direct effect was found to enhance moisture over NE India, whereas indirect effect was found to reduce surface moisture as well as atmospheric stability and reduce rainfall. Hence, the direct effect was primarily responsible for rainfall enhancement.

Changes in manuscript: Spatial distribution of CCN has been added in Figure 11 which show enhancement over NE India.